# Genomic characterization of foodborne *Salmonella enterica* and *Escherichia coli* isolates from Saboba district and Bolgatanga Municipality Ghana

Gabriel Temitope Sunmonu[1], Frederick Adzitey[2], Erkison Ewomazino Odih[1], Boniface Awini Tibile[2], Rejoice Ekli[2], Martin Aduah[2], Anderson O. Oaikhena[1], Olabisi C. Akinlabi[1], Akebe Luther King Abia[3], Daniel Gyamfi Amoako[3,4], Iruka N. Okeke[1] *

1 Department of Pharmaceutical Microbiology, Faculty of Pharmacy, University of Ibadan, Ibadan, Nigeria, 2 Faculty of Agriculture, Food and Consumer Sciences, University for Development Studies, Tamale, Ghana, 3 Antimicrobial Research Unit, College of Health Sciences, University of KwaZulu-Natal, Durban, South Africa, 4 Department of Integrative Biology and Bioinformatics, University of Guelph, Guelph, Ontario, Canada

* iruka.n.okeke@gmail.com

**Data Availability Statement:** The sequence reads were deposited in the European Nucleotide Archive with the study accession PRJEB58695 (For ease of

## Abstract

*Salmonella enterica* and *Escherichia coli* are well-known bacteria commonly associated with foodborne illnesses in humans and animals. Genomic characterization of these pathogens provides valuable insights into their evolution, virulence factors and resistance determinants. This study aimed to characterized previously isolated *Salmonella* (n = 14) and *E. coli* (n = 19) from milk, meat and its associated utensils in Ghana using whole-genome sequencing. Most of the *Salmonella* serovars (Fresno, Plymouth, Infantis, Give and Orleans) identified in this study are yet to be reported in Ghana. Most *Salmonella* isolates were pan-sensitive, but genes conferring resistance to fosfomycin (*fosA7.2*) and tetracycline (*tet (A)*) were detected in one and three isolates, respectively. Seven of the *Salmonella* isolates carried the IncI1-I(Gamma) plasmid replicon. Although antimicrobial resistance was not common among *Salmonella* strains, most (11/19) of the *E. coli* strains had at least one resistance gene, with nearly half (8/19) being multidrug resistant and carried plasmids. Three of the 19 *E. coli* strains belonged to serovars commonly associated with enteroaggregative *E. coli* (EAEC) pathotype. While strains belonging to virulence-associated lineages lacked key plasmid-encoded virulence plasmids, several plasmid replicons were detected in most of the *E. coli* (14/19) strains. Food contaminated with these pathogens can serve as a vehicle for disease transmission, posing a significant public health risk and necessitating stringent food safety and hygiene practices to prevent outbreaks. Hence, there is need for continuous surveillance and preventive measures to stop the spread of foodborne pathogens and reduce the risk of associated illnesses in Ghana.

retrieval, a full listing of the specific sequences is included in S1 Table).

**Funding:** Whole genome sequencing and analysis for this project was supported through SEQAFRICA, a regional award funded by the Department of Health and Social Care's Fleming Fund using UK aid. INO is a Calestous Juma Fellow supported by the Bill and Melinda Gates Foundation (INV-036234). The views expressed in this publication are those of the authors and not necessarily those of the Bill and Melinda Gates Foundation, the UK Department of Health and Social Care or its Management Agent, Mott MacDonald. The funders had no role in study design, data collection and analysis, decision to publish, or preparation of the manuscript.

**Competing interests:** The authors have declared that no competing interests exist.

## Introduction

Milk and meat are essential protein sources that constitute a significant and nutrient-rich component of human diets. However, their consumption is often associated with foodborne infections [1], particularly those caused by *Salmonella* and *Escherichia coli* [2–5]. Salmonellosis is under-reported in Ghana, and only a few studies have investigated the plausible role of contaminated milk [6], meat, meat products, handlers' hands and associated surfaces such as knives, tables and aprons [7–9] in facilitating their transmission. Knowledge of food safety practices by key food handlers in Ghana has recently been reported to be suboptimal, as have food safety infrastructure and regulatory enforcement [10, 11]. There have been a few reports of meat samples contaminated with *E. coli* in Ghana [11–14]. Similarly, *E. coli* has been recovered from milk, milking utensils, faeces of lactating cow and milkers' hands [15]. Some of these *E. coli* strains harbour both virulence and antimicrobial resistance genes (ARGs), raising public health concerns. However, few of these strains have been thoroughly characterized.

There are public health and food safety implications of finding *Salmonella* and *E. coli* in food because they are invariably of faecal origin. Globally, in addition to contamination risks, *Salmonella* and *E. coli* sourced from milk and meat increasingly exhibit resistance to different classes of antibiotics that is commonly mediated by mobile elements [13, 16–18]. The prevalence of multidrug-resistant (MDR) *Salmonella* and *E. coli* is also on the rise in clinical infections [19, 20]. Notably, resistance to extended spectrum beta-lactams, trimethoprim/sulfamethoxazole, chloramphenicol and ciprofloxacin has been reported, often associated with plasmids that could mediate their spread, in both *Salmonella* and *E. coli* isolates from milk and retail meats in Ghana [6, 12, 14].

Various methods, including serotyping, antibiotic profiling, pulsed-field gel electrophoresis and whole genome sequencing, have been employed to elucidate the phenotypic and genotypic attributes of foodborne pathogens and to determine their interrelationships and connections to pandemic clones of interest [13, 21]. Next-generation sequencing (NGS) technology, the most versatile and informative approach, has gained recent prominence [22, 23]. NGS is now used by PulseNet to categorize foodborne diseases, enabling nuanced epidemiological investigations [24]. Identification of virulence factors, antibiotic resistance genes, and serotypes are all possible by genomic analysis, which can also provide enhanced information on strain interrelatedness, and therefore enable source attribution. Data on the genomic characterization of *Salmonella* and *E. coli* from milk and meat are few from low- and middle-income countries, including Ghana [7, 25]. In light of this gap, this study aims to characterize the resistance, virulence and plasmid profile of previously isolated *Salmonella* and *E. coli* isolated from fresh different retail meats, milk, and associated samples (handler's hand swab, table, knife and faecal samples) in Saboba district and Bolgatanga Municipality of Ghana.

## Methods

### Strains

A total of 33 bacterial isolates (14 *Salmonella* and 19 *E. coli* species) previously isolated from various fresh and ready-to-eat meats, meat sellers' tables, milk, milk-collecting utensils, milkers' hands and faeces of lactating cows were characterized for this study (Table 1) [8, 15, 26, 27]. The isolates originated from markets and farms in Bolgatanga Municipality and Saboba District in Northern Ghana and were cryopreserved in 50% glycerol in Luria broth at -80˚C.

### Ethical considerations

All isolates were recovered in earlier studies from vended food or at informal food vending premises, including milk cow droppings [8, 15, 26, 27]. Study design and sampling was

**Table 1.  *E. coli* and *Salmonella* isolates characterized in this study.**

| Id | Code | Collection date | Species | Sample | Reference |
|---|---|---|---|---|---|
| GH-FA-M23_S31 | M2 3 | 2/10/2020 | *Salmonella enterica* | Milk | [27] |
| GH-FA-M87_S6 | M8 1 | 2/10/2020 | *Salmonella enterica* | Milk | [27] |
| GH-FA-FS24_S23 | FS24 | 2/10/2020 | *Salmonella enterica* | Faecal sample (Milking cow) | [27] |
| GH-FA-M25_S2 | M25 | 2/10/2020 | *Salmonella enterica* | Milk | [27] |
| GH-FA-US15_S32 | US15-2 | 22/10/2020 | *Salmonella enterica* | Utensil sample | [27] |
| GH-FA-FK4_S10 | FK4 | 22/10/2020 | *Salmonella enterica* | Knife (Fresh meat) | [8] |
| GH-FA-FK4d_S7 | FK4d | 22/10/2020 | *Salmonella enterica* | Knife (Fresh meat) | [8] |
| GH-FA-RCH2_S11 | Rch2 extra | 22/10/2020 | *Salmonella enterica* | RTE Chicken | [8] |
| GH-FA-RG2_S29 | RG2 | 22/10/2020 | *Salmonella enterica* | RTE Guinea fowl | [8] |
| GH-FA-RK4_S19 | RK4 | 2/10/2020 | *Salmonella enterica* | Knife (RTE utensil) | [8] |
| GH-FA-RM4_S20 | RM4 | 2/10/2020 | *Salmonella enterica* | RTE Mutton | [8] |
| GH-FA-RP5_S5 | RP5 | 12/10/2020 | *Salmonella enterica* | RTE Pork | [8] |
| GH-FA-RP5D_S11 | RP5D | 12/10/2020 | *Salmonella enterica* | RTE Pork | [8] |
| GH-FA-RPSD_S30 | RPSD | 12/10/2020 | *Salmonella enterica* | RTE Pork | [8] |
| GH-FA-FS23_S22 | FS23 | 22/10/2020 | *Escherichia coli* | Faecal sample (milking cow) | [27] |
| GH-FA-US24_S14 | US24 | 22/10/2020 | *Escherichia coli* | Utensil Sample | [15] |
| GH-FA-HS11_S26 | HS11 | 22/10/2020 | *Escherichia coli* | Hand swab | [15] |
| GH-FA-HS3_S27 | HS3 | 2/10/2020 | *Escherichia coli* | Hand swab | [15] |
| GH-FA-M9_S16 | M9 | 22/10/2020 | *Escherichia coli* | Milk | [15] |
| GH-FA-US3_S21 | US3 | 2/10/2020 | *Escherichia coli* | Utensil sample | [15] |
| GH-FA-HS12_S21 | HS12 | 2/10/2020 | *Escherichia coli* | Hand Swab | [15] |
| GH-FA-M25D_S22 | M25 | 2/10/2020 | *Escherichia coli* | Milk | [15] |
| GH-FA-FB4_S24 | FB4 | 22/10/2020 | *Escherichia coli* | Fresh Beef | [26] |
| GH-FA-FC1_S23 | FC1 | 2/10/2020 | *Escherichia coli* | Fresh Chicken | [26] |
| GH-FA-FH2_S15 | FH2 | 22/10/2020 | *Escherichia coli* | Hand Swab (Fresh meat vendor) | [26] |
| GH-FA-FGS_S16 | FG5 | 22/10/2020 | *Escherichia coli* | Fresh Guinea fowl | [26] |
| GH-FA-FM4_S34 | FM4 | 12/10/2020 | *Escherichia coli* | Fresh Mutton | [26] |
| GH-FA-FT1_S17 | FT1 | 12/10/2020 | *Escherichia coli* | Table swab (Fresh meat) | [26] |
| GH-FA-RB4_S24 | RB4 | 12/10/2020 | *Escherichia coli* | RTE Beef | [26] |
| GH-FA-RH4_S3 | RH4 | 12/10/2020 | *Escherichia coli* | Hand swab (RTE meat Vendor) | [26] |
| GH-FA-RT3_S6 | RT3 | 12/10/2020 | *Escherichia coli* | Table swab (RTE meat) | [26] |
| GH-FA-RT3_S29 | RT3d | 12/10/2020 | *Escherichia coli* | Table swab (RTE meat) | [26] |
| GH-FA-RCLI_S18 | Rch1 | 12/10/2020 | *Escherichia coli* | RTE Chevon | [26] |

Key: RTE: Ready-to-eat–meats sampled in prepared form

approved by the Department of Veterinary Science, UDS. No other permissions were obtained or deemed necessary by the department. No humans or animals were use in the research and therefore ethical approval was deemed not required.

### *Salmonella* and *E. coli* identification

*Salmonella* isolates were initially confirmed using a latex agglutination kit for *Salmonella* (Oxoid Limited, Basingstoke, UK) and by PCR targeting the *invA* gene as described by Rahn et al. (1992) [28], using PCR oligonucleotides *invA139f* GTGAAATTATCGCCACGTTCGGGCAA and *invA141r* TCATCGCACCGTCAAGGAACC. PCR was performed using PuRe Taq Ready-To-Go PCR Beads (illustra™). The PCR cycle used an initial denaturation temperature of 95˚C for two minutes, followed by 35 cycles of denaturation at 95˚C for 30 seconds, annealing

at 55˚C for 30 seconds and extension at 72˚C for two minutes, then a terminal extension at 72˚C for five minutes. Visualization of the 284 bp amplicon was accomplished after electrophoresis on 1.5% (w/v) agarose gels stained with Gel red (biotium), using a UVP GelMax transilluminator and imager. *Salmonella* isolates positive for *invA* and *E. coli* isolates were biotyped with the Gram-negative (GN) test kit (Ref: 21341) on VITEK 2 systems (version 2.0, Marcy-l'Etoile, France, Biomérieux) according to manufacturer's instructions

## DNA extraction, library preparation and whole genome sequencing

Genomic DNA of the isolates was extracted using Wizard DNA extraction kit (Promega; Wisconsin, USA) in accordance with manufacturer's protocol. Using a dsDNA Broad Range quantification assay, the extracted DNA was quantified on a Qubit fluorometer (Invitrogen; California, USA). dsDNA libraries were prepared using NEBNext Ultra II FS DNA library kit for Illumina with 96-unique indexes (New England Biolabs, Massachusetts, USA; Cat. No: E6609L). DNA libraries was quantified using dsDNA High Sensitivity quantification assay on a Quibit fluorometer (Invitrogen; California, USA) and fragment length analysed with the Bioanalyzer (Agilent). Denatured libraries were sequenced on an Illumina MiSeq (Illumina, California, USA). The raw sequence reads were *de novo* assembled using SPAdes v3.15.3 [29] according to GHRU protocols (https://gitlab.com/cgps/ghru/pipelines/dsl2/pipelines/assembly).

## Sequence typing of *Salmonella* and *E. coli* genomes

Sequence reads were deposited in the *Salmonella* and *E. coli* database for *Salmonella* and *E. coli* respectively on EnteroBase [30] and analyzed using publicly available tools that we have previously validated [31, 32]. Multi-locus sequence types (MLST) for the isolates were determined using ARIBA [33]. Novel ST strains were assigned ST using EnteroBase [30]. The *Salmonella* genome assemblies were analysed using the *Salmonella* In-Silico Typing Resource (SISTR) for the prediction of serovars and serogroups [34] (https://github.com/phac-nml/sistr_cmd), while the serotyping of the *E. coli* genome was done using ECtyper [35].

## Identification of AMR, plasmids and virulence genes

PlasmidFinder [36] was utilized to identify plasmid replicons that were present in the assembled genomes. AMRFinderPlus v3.10.24 [37] and its associated database (version 2022-04-04.1) were used to predict the antimicrobial resistance genes carried by the isolates and the drug classes to which they probably conferred resistance. Using ARIBA [33] and the virulence factor database (VFDB, http://www.mgc.ac.cn/VFs/), we were also able to identify the virulence genes that were present in the isolates.

## Single Nucleotide Polymorphism (SNP) calling and phylogenetic analysis

For phylogenetic analysis, reference sequences for the *Salmonella* and *E. coli* genomes were objectively selected from the National Center for Biotechnology Information Reference Sequence (RefSeq) database (https://www.ncbi.nlm.nih.gov/refseq/) using BactinspectorMax v0.1.3 (https://gitlab.com/antunderwood/bactinspector). The selected references were the *S. enterica* subsp. enterica serovar Fresno strain (assembly accession: GCF_003590695.1) and the *E. coli* O25b:H4-ST131 strain (assembly accession: GCF_000285655.3). The sequence reads for each species were then mapped to the chromosome of the reference using BWA (v0.7.17) [38] and variants were called and filtered using bcftools (v1.9) [39] as implemented in the GHRU SNP phylogeny pipeline (https://gitlab.com/cgps/ghru/pipelines/snp_phylogeny). Variant

positions were concatenated into a pseudoalignment and used to generate a maximum likelihood tree using iqtree (v1.6.8) [40]. SNP distances between the genome pairs were calculated using snp-dists v.0.8.2 (https://github.com/tseemann/snp-dists).

# Results

## *Salmonella* serotypes, sequence types (STs), virulence factors and phylogeny

We used SISTR software to predict the serovars of the 14 *Salmonella* strains characterized in this study from whole genome sequence reads, which are deposited in the European Nucleotide Archive with the study accession PRJEB58695 (S1 Table). The most common serotype was Fresno (n = 6) followed by Give (n = 3), Orleans (n = 2), Plymouth (n = 1), Agona (n = 1) and Infantis (n = 1). The *S.* Fresno and *S.* Orleans isolates were from previously unreported sequence types, now designated ST10742 and ST10465 respectively (S2 Table). As shown in Fig 1, all the isolates from ready-to-eat pork, mutton and chicken belonged to serovar Fresno. *S.* Fresno isolates were also isolated from a meat vendor's knife, as was *S.* Orleans. The three milk isolates, which were all from Saboba, belonged to the serovars Plymouth (ST565), Give (ST516) and Agona (ST13) (Fig 1). Two more ST516 *S.* Give isolates were recovered from the faeces of milking cow and from a milking utensil.

Phylogenetic analysis of the 14 *Salmonella* isolates from this study showed that all *S.* Fresno isolates, irrespective of source, clustered together and differed by < 3 SNPs. The two *S.* Orleans isolates were identical (0 SNPs) and the three *S.* Give isolates were also identical, with the isolates originating from milk, faeces of milking cow and from a milking utensil, also in Saboba.

All *Salmonella* isolates harboured curli (*csg*) genes as well as *bcf*, *fim* and *ste* fimbrial operons and ten of them, representing all serovars except *S.* Give and Plymouth, carried long polar fimbriae (*lpf*) genes. The *S.* Infantis and *S.* Agona strains carried *ratB* and *shdA*. Type III secretion system effector genes, such as: *inv*, *org*, *prg*, *sif*, *spa*, *sse*, *ssa* and *sop* were detected in all the

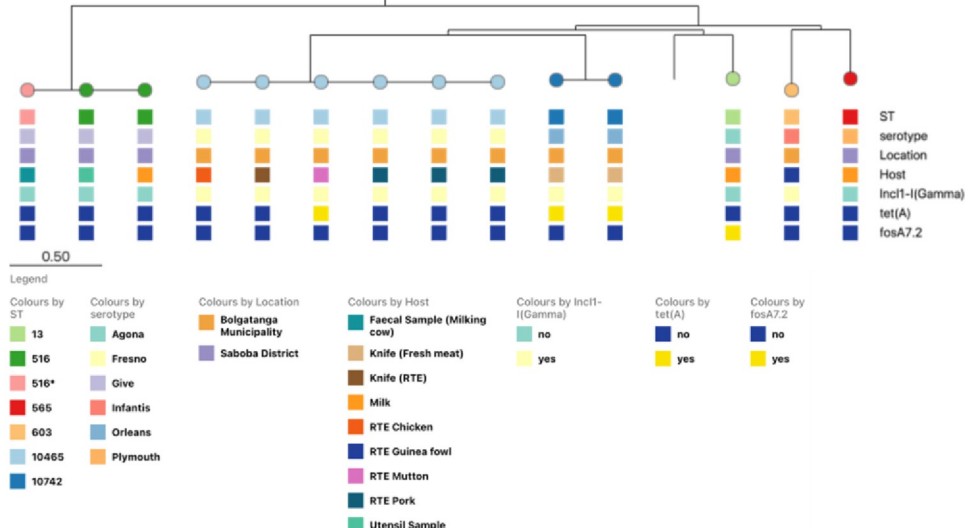

**Fig 1. Core genome SNP-based maximum likelihood tree showing phylogenetic relationships among strains sequenced in this study.** The matrix below the tree shows the sequence types, serotypes, location of isolation (Bolgatanga municipality or Saboba district) as well as the source/host sample and presence or absence of the IncI1-I (Gamma) plasmid replicon, *tetA* and *fosA* resistance genes.

isolates while *avr* was present in 57.1% (8/14) of the isolates and only one isolate harboured *gogB*. Four of the isolates also encoded the cytholethal distending toxin gene, *cdtB* (S3 Table).

## Plasmid replicons and ARG profiles of *Salmonella*

The *Salmonella* isolates were largely pan-sensitive but genes conferring resistance to fosfomycin (*fosA7.2*) and tetracycline (*tet(A)*) were detected in one and three isolates respectively. Both *S*. Orleans isolates and one of the *S*. Fresno, from ready-to-eat mutton carried *tetA*, along with an IncI1-I(Gamma) plasmid replicon, which was also seen in six other *tetA*-negative strains. Interestingly, the IncI1-I(Gamma) plasmid replicon was detected in all isolates from Bolgatanga municipality, irrespective of serovar, and no isolate from Saboba district harboured this plasmid. The fosfomycin resistance gene was found in the *S*. Agona genome, in which no plasmid replicons were detected (Fig 1).

## *E. coli* serotypes, sequence types (STs), virulence factors and phylogeny

A total of 19 *E. coli* isolates were identified. *E. coli* serotyping using the ECtyper revealed that the most common serotypes among the *E. coli* isolates were -:H7 (n = 2), O138:H48 (n = 2), O6:H16 (n = 2), and O8/O160:H16 (n = 2). A number of these serovars and STs are associated with pathogenicity, notably O6:H16 [41], as well as O8/O160:H16 and O77/O17/O44/O106/O73:H18 (ST394; [42–44]). The strains belonging to these lineages lacked the defining virulence genes of the respective pathotypes but did contain accessory virulence genes, as shown in S4 Table.

Irrespective of whether they belonged to a lineage commonly associated with virulence, most of the isolates contained a range of adhesins and iron utilization genes. *E. coli* extracellular protein (ECP) export pathway (*ecp/yag*) and *ompA* and type I fimbriae-encoding operon, *fim*, encoding genes seen in most *E. coli* genomes, were present in 94.7% (18/19) of *E. coli* isolates. Fimbriae encoding gene, *f17d*, often seen in enterotoxigenic *E. coli* recovered from animals, was present in the two O8/O160:H16 isolates and an O-:H7 isolate.

The phylogenetic analysis of the 19 *E. coli* isolates from this study and a reference genome (NZ_HG941718.1) based on SNP is presented in Fig 2. The range of isolates was broader than with *Salmonella* but closely related pairs of isolates belonging to the same serovar, and ST were found in three instances. Very similar (2347 SNPs) O6:H16 isolates were recovered from different food preparation table samples in Bolgatanga. One of the two isolates from milk in Saboba belonged to ST2165 and was identical (0 SNPs) to a Saboba ST2165 utensil isolate. The two ST4 isolates from fresh beef and a cow milker's hand differed by 2347 SNPs and are unlikely to be connected.

## Plasmid replicons and ARG profiles of *E. coli*

Antimicrobial resistance determinants present in the *E. coli* isolates include those encoding resistance to aminoglycosides (*aph(3")-Ib*, *aph(6)-Id*, *aph(6)-Id*, *aph(3")-Ib*), beta-lactams (*bla*$_{LAP-2}$, *bla*$_{TEM-1}$), fosfomycin (*fosA7.5*), quinolones (*qnrB19*, *qnrS1*), sulfonamide (*sul2*), tetracycline (*tet(A)*, *tet(B)*) and trimethoprim (*dfrA14*) (Fig 2). At least 3 antimicrobial resistance genes (ARGs) which confer resistance to different classes of antibiotics were present in 8 isolates. Three isolates carried one ARG each while 8 isolates had no ARGs. Six strains carried the genes *aph(3")-Ib*, *aph(6)-Id*, *blaTEM-1*, *dfrA14*, *sul2*, *qnrS1* and *tet(A)*. The *dfrA14-qnrS1-tet (A) resistance* gene combination has previously been reported from Nigeria, being part of a transposon transmitted in an IncX plasmid [45]. In this study however, IncX replicons were not detected. The most common plasmid replicon type detected among the *E. coli* isolates was pO111 (n = 6), originally described in an *E. coli* virulence plasmid and found in the

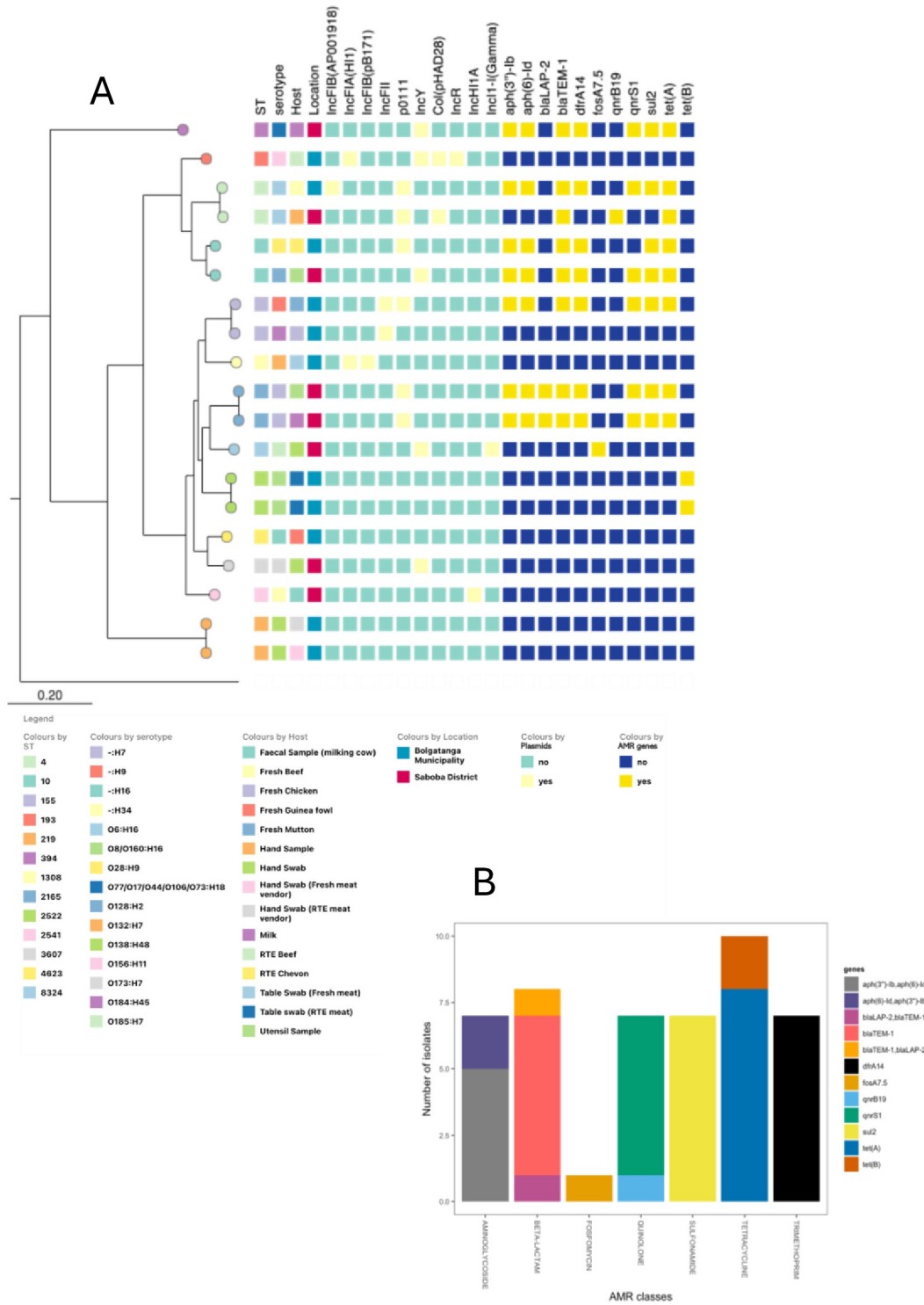

**Fig 2. Phylogeny, plasmid replicons and resistance genes of *E. coli* isolates.** (a). SNP-based phylogenetic tree of the *E. coli* isolates showing the range of sequence types, serovars, plasmid replicons and resistance genes detected (b) AMR determinants detected in *E. coli* and the resistance they confer.

aforementioned strains belonging to virulence-associated lineages. The other plasmid replicon types detected were IncY (n = 5), IncFII (n = 2), IncFIA(HI1) (n = 2), Col(pHAD28) (n = 2), IncFIB(pB171) (n = 1), IncR (n = 1), IncHI1A (n = 1) and IncI1-I(Gamma), which was common among the *Salmonella*, (n = 1) (Fig 2). All multidrug resistant *E. coli* strains in this study encode pO111 or IncY replicons.

## Discussion

*Salmonella* and *E. coli* are the main causes of bacterial foodborne illnesses in Ghana [46]. Retail meat, along with milk and their products are recognized as primary sources of foodborne Salmonellosis [12] and *E. coli* infection [7]. Post-cooking handling practices, exposure during the points of sale, and suboptimal meat storage conditions collectively contribute to an increased presence of both pathogenic and spoilage bacteria in ready-to-eat (RTE) meat [47]. Within Ghana, food safety has been inadequately studied in the northern region [10]. In this study, we characterized the genomes of 14 *Salmonella* and 19 *E. coli* previously isolated from Saboba district and Bolgatanga Municipality in northern Ghana.

The *Salmonella* serovars (Fresno, Plymouth, Infantis, Give and Orleans) identified in this study are yet to be reported in Ghana, but Guinee et al. (1961) [48] isolated *S*. Agona from cattle in Ghana and *S*. Give (but ST524, different from ST516 in this study) has been reported from beef in Nigeria [49]. Isolation of *S*. Infantis from retail poultry meat has also been reported in Ecuador [50], Belgium [51] and Italy [52] and all the serovars detected in this study have been implicated in human infections. The identification of *Salmonella* serovars not previously documented in the country in meat and milk products highlights the need for heightened surveillance and preventive measures to curb the spread of foodborne pathogens and reduce the risk of associated illnesses.

Unlike *Salmonella*, not all *E. coli* are potential pathogens. However, *E. coli* serve as markers for faecal contamination and therefore the potential that other pathogens are present. The predominant *E. coli* STs (ST4, ST10, ST219, ST2522) detected in our study have been previously isolated from food animals and have been associated with pathogenicity [53–55].

While none of the *E. coli* isolates carried genes encoding ETEC heat-sensitive or heat-labile enterotoxins, f17d fimbrial genes present in four of the *E. coli* genomes encode ETEC colonization factors commonly associated with colonization of cattle and other ruminant isolates [56]. *E. coli*, F17 fimbriae are associated with pathogenic *E. coli* recovered from diarrhoea and septicaemia outbreaks in calves, lambs, and humans, including from outbreaks.

Additionally, two of the ST4 *E. coli* isolates from this study not harbouring f17d fimbrial genes belong to the serovar O6:H16, one of the most widely disseminated lineages of human enterotoxigenic *E. coli* (ETEC). O6:H16 ETEC cause outbreaks, often associated with food and/or inadequate handwashing [57–59]. ETEC, by definition, produce plasmid-encoded heat-labile and/or heat stable toxins not present in the genomes of the isolates from this study.

However, the serovars O8/O160:H16 and O77/O17/O44/O106/O73:H18 belong to a previously described enteroaggregative *E. coli* (EAEC) lineage [42, 44]. Isolates from this study belonging to the serovars O8/O160:H16 and O77/O17/O44/O106/O73:H18 possessed no EAEC accessory genes. These strains are from virulent lineages but lack key virulence genes that are plasmid-encoded, which could mean that these plasmids were lost in the food chain or during isolation but could be reacquired. Nevertheless, the presence of these strains in food could increase the risk of foodborne illness.

While strains belonging to virulence-associated lineages lacked key plasmid-encoded virulence plasmids, several plasmid replicons were detected in the isolate genomes. According to McMillan et al. (2019) [60], plasmid replicons ColE, IncI1, IncF, and IncX were commonly

detected in *Salmonella* from food animals in the US. In this study, the IncI1 replicon was predominant, with nine of the thirteen *Salmonella* strains harbouring IncI1 plasmid replicon, of which three harboured the *tetA* gene. This is likely to be an instance of a successful mobile element with extraordinary local reach, a few of which have been reported from West Africa, including Ghana, in the past [45, 61–63]. The IncI1-I(Gamma) plasmid replicon observed among *Salmonella* isolates was detected in all *Salmonella* isolates from Bolgatanga municipality—three different serovars—and none of the Saboba district isolates. However, one *E. coli* isolate (from a recently reported ST, ST8274) from Saboba did have this replicon. As it is an IncI1 plasmid replicon, its plasmid should be better characterized and, it should remain under surveillance because numerous articles have reported association of the IncI1 plasmid replicon with multiple ARGs, such as *tetB*, *tetAR*, $bla_{CMY-2}$, $bla_{TEM-1}$, *aac3VIa*, *aphA*, *aadA* and *sul1* [60], $bla_{CTXM-1}$ [64], *strA*, *strB*, *cmlA*, *floR*, $bla_{SHV-12}$, $bla_{OXA-2}$ and *FosA3* [65] in IncI1 plasmids in *Salmonella*. As our own sequence was generated by short read only, the first step would be to generate long read sequence that could fully assemble the plasmid and make it possible to identify genetic factors supporting its success.

Among the *E. coli* isolates, the plasmid replicons pO111 was the most common replicon. A previous study by Balbuena-Alonso et al. (2022) [66], revealed that pO111 is usually associated with extended spectrum beta lactamases gene and is very common in food and clinical isolates. In this study, all isolates carrying pO111 harbour at least one beta-lactamase gene. Likewise, all the pO111 plasmid bearing isolates in this study carried ARGs that confer resistance to at least 4 classes of antibiotics. Altogether, these data demonstrate a concerning reservoir of resistance genes in these foodborne bacteria.

## Conclusion

This study has characterized the genomes of *Salmonella* and *E. coli* in milk, meat and their associated utensils. The diverse serovars and virulence genes detected in *Salmonella* strains indicate potential pathogenicity. Although not all *E. coli* strains are pathogenic, their presence serves as an indicator of faecal contamination, suggesting the potential presence of other harmful pathogens. The presence of EAEC strains in food is concerning as EAEC is a well-known cause of diarrhoeal diseases, particularly in children and immunocompromised individuals, making its presence in food a serious concern. While antimicrobial resistance was not common among *Salmonella* strains, most of the *E. coli* strain had at least one resistance gene, and almost half were multidrug resistant and carried mobile elements. Moreover, there have been recent reports of resistant *Salmonella* and E. coli from meat and milk elsewhere in West Africa [49].

A recent scoping review reported weak enforcement of food safety regulations, as well as a lack of infrastructure, knowledge and skills to implement these regulations [10, 11, 46]. Food contaminated with and *Salmonella* and *E. coli* can serve as a vehicle for their transmission, posing a significant public health risk. We recommend that food safety regulations be strengthened in northern Ghana and, by extension, West Africa. It is also important to increase awareness among consumers so that food is handled in such a way to prevent pathogen transmission. There is an additional need for continuous surveillance and preventive measures to stop the spread of foodborne pathogens and reduce the risk of associated illnesses in Ghana.

## Supporting information

**S1 Table. Accession numbers for genomes generated in this study.**
(XLSX)

**S2 Table. Novel *Salmonella* allelic profile and assigned ST.**
(XLSX)

**S3 Table. *Salmonella* strain metadata including serotype, ST, plasmid replicon, AMR and virulence profile.**
(XLSX)

**S4 Table. *E. coli* strain metadata including serotype, ST, plasmid replicon, AMR and virulence profile.**
(XLSX)

## Acknowledgments

We thank Ayorinde Afolayan, Faith I. Oni, and Abeeb Adeniyi for technical assistance and Jola-Ade Ajiboye, Kesiena Akpede and Pernille Nilsson for logistic support.

## Author Contributions

**Conceptualization:** Frederick Adzitey, Iruka N. Okeke.

**Data curation:** Gabriel Temitope Sunmonu, Frederick Adzitey, Erkison Ewomazino Odih, Anderson O. Oaikhena, Olabisi C. Akinlabi, Iruka N. Okeke.

**Formal analysis:** Gabriel Temitope Sunmonu, Erkison Ewomazino Odih, Olabisi C. Akinlabi.

**Funding acquisition:** Iruka N. Okeke.

**Investigation:** Gabriel Temitope Sunmonu, Erkison Ewomazino Odih, Boniface Awini Tibile, Rejoice Ekli, Martin Aduah, Anderson O. Oaikhena, Olabisi C. Akinlabi.

**Methodology:** Erkison Ewomazino Odih, Anderson O. Oaikhena, Iruka N. Okeke.

**Project administration:** Anderson O. Oaikhena, Iruka N. Okeke.

**Resources:** Iruka N. Okeke.

**Supervision:** Frederick Adzitey, Anderson O. Oaikhena, Akebe Luther King Abia, Daniel Gyamfi Amoako, Iruka N. Okeke.

**Validation:** Erkison Ewomazino Odih, Anderson O. Oaikhena, Iruka N. Okeke.

**Visualization:** Gabriel Temitope Sunmonu, Erkison Ewomazino Odih.

**Writing – original draft:** Gabriel Temitope Sunmonu, Erkison Ewomazino Odih, Iruka N. Okeke.

**Writing – review & editing:** Gabriel Temitope Sunmonu, Frederick Adzitey, Erkison Ewomazino Odih, Boniface Awini Tibile, Rejoice Ekli, Martin Aduah, Anderson O. Oaikhena, Olabisi C. Akinlabi, Akebe Luther King Abia, Daniel Gyamfi Amoako, Iruka N. Okeke.

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
