## [Decision Letter · Decision Letter 0]

1 Oct 2024

PONE-D-24-40234Genomic characterization of foodborne Salmonella enterica and Escherichia coli isolates from Saboba district and Bolgatanga Municipality GhanaPLOS ONE

Dear Dr. Okeke,

Thank you for submitting your manuscript to PLOS ONE. After careful consideration, we feel that it has merit but does not fully meet PLOS ONE’s publication criteria as it currently stands. Therefore, we invite you to submit a revised version of the manuscript that addresses the points raised during the review process.

We look forward to receiving your revised manuscript.

Kind regards,

Mabel Kamweli Aworh, DVM, MPH, PhD. FCVSN

Academic Editor

PLOS ONE

Journal Requirements:

3. Thank you for submitting the above manuscript to PLOS ONE. During our internal evaluation of the manuscript, we found significant text overlap between your submission and previous work in the [introduction, conclusion, etc.].

Please revise the manuscript to rephrase the duplicated text, cite your sources, and provide details as to how the current manuscript advances on previous work. Please note that further consideration is dependent on the submission of a manuscript that addresses these concerns about the overlap in text with published work.

[If the overlap is with the authors’ own works: Moreover, upon submission, authors must confirm that the manuscript, or any related manuscript, is not currently under consideration or accepted elsewhere. If related work has been submitted to PLOS ONE or elsewhere, authors must include a copy with the submitted article. Reviewers will be asked to comment on the overlap between related submissions (http://journals.plos.org/plosone/s/submission-guidelines#loc-related-manuscripts).]

We will carefully review your manuscript upon resubmission and further consideration of the manuscript is dependent on the text overlap being addressed in full. Please ensure that your revision is thorough as failure to address the concerns to our satisfaction may result in your submission not being considered further.

“Whole genome sequencing and analysis for this project was supported through SEQAFRICA award to INO. The SEQAFRICA project is funded by the Department of Health and Social Care’s Fleming Fund using UK aid. The views expressed in this publication are those of the authors and not necessarily those of the UK Department of Health and Social Care or its Management Agent, Mott MacDonald. INO is a Calestous Juma Fellow supported by the Bill and Melinda Gates Foundation.”

Additional Editor Comments:

In addition to your response to the reviewer's comments, please address the following;

**1a. Clarify the Use of Previously Isolated Bacteria in the Study**

It would be helpful if the abstract and aim mention that the study will be using previously isolated bacteria. This is a critical detail that should be highlighted early on for clarity and to set appropriate expectations for the reader.

**b. Address the Discrepancies in Sampling Methods**

Upon reviewing references 22 and 23, the sampling methodologies differ from those described in this study. Specifically:Neither reference mentions the collection of fresh meat or milk samples.The term *lactating cows* does not appear in the methods or anywhere else in these papers.There is no mention of samples from handlers’ hands, meat sellers’ tables, knives, or feces in these papers.The sampling procedures of the two references are quite similar in terms of Ready-to-Eat (RTE) sample collection, yet there is no mention of fresh meat, milk, or surfaces like tables or utensils in those papers.
**Suggestions to Address These Points:**Provide a clear explanation of how the current study's sampling methods relate to the methods in references 22 and 23.Clarify if the fresh meat, milk, and surface samples mentioned in your manuscript are new, or if they are derived from RTEs as per references 22 and 23.If there is an overlap or distinction between RTE and fresh meat sampling, this should be explicitly discussed in the text. For example, is fresh meat being classified as RTE in some instances? If so, it would be important to explain this decision clearly to avoid confusion.

**c. Clarification on the Synonymity Between RTEs and Fresh Meat**

If the authors are suggesting that Ready-to-Eat foods (RTEs) are synonymous with fresh meat or that there is some overlap between the two, it is crucial to state this explicitly. Readers may assume a difference between the two categories, and any potential synonymity should be justified with an appropriate explanation.

**d. Enhance transparency in sampling descriptions and briefly describe how these bacteria were isolated from these samples.**

Consider revising the methods section to provide brief description about how *E. coli *and *Salmonella* were isolated from the fresh meat, milk, and RTE samples. Also describe the sampling procedures for fresh meat, milk and other samples if they were conducted separately from the RTE samples. This will help clarify any misconceptions and make it clear how the study builds on or differs from the previously published studies.

By addressing these points, the study will offer greater transparency in its methodology and improve clarity for readers regarding the sampling strategy and its connection to previously published studies. 

**2. Highlight key limitations of the present study.**

Reviewers' comments:

Reviewer's Responses to Questions

**Comments to the Author**

1. Is the manuscript technically sound, and do the data support the conclusions?

Reviewer #1: Partly

Reviewer #2: Yes

Reviewer #3: Partly

Reviewer #4: Yes

2. Has the statistical analysis been performed appropriately and rigorously? 

Reviewer #1: N/A

Reviewer #2: Yes

Reviewer #3: N/A

Reviewer #4: N/A

3. Have the authors made all data underlying the findings in their manuscript fully available?

Reviewer #1: Yes

Reviewer #2: Yes

Reviewer #3: Yes

Reviewer #4: Yes

4. Is the manuscript presented in an intelligible fashion and written in standard English?

Reviewer #1: Yes

Reviewer #2: Yes

Reviewer #3: Yes

Reviewer #4: Yes

5. Review Comments to the Author

Reviewer #1: Abstract: Review and either complete the half sentences and remove where not applicable. Eg...."While 7of the Salmonella isolates carry the IncI1-I(Gamma) plasmid replicon."

Review the recommendation and make it more focused on the work and not broad, blanket recommendation.

Introduction:

The role of the equipment such knives and surfaces need to be mentioned in the introduction and not just having it appear in the objectives.

Method:

A lot is described for Salmonella but not much is mentioned about how the E.coli samples were presented.

Results and Discussion:

I was expecting the authors to emphasize the genetic relatedness for the Salmonella and E. coli and their implication on foodborne illnesses and and control.

Other details are available in the body of the manuscript.

Overall this is an excellent work and the presentation is nicely done.

Reviewer #2: The manuscript is technically sound, with experiments conducted rigorously, including appropriate controls, replication, and sample sizes. The conclusions are well-supported by the data presented, without any overreach or misinterpretation. The statistical analysis has been performed appropriately and rigorously, ensuring that the findings are valid and reliable. The manuscript is also presented in an intelligible fashion, written in clear and standard English, making it easy to follow and understand for the intended scientific audience.

Reviewer #3: Undoubtedly, this study is of value from a public health and One Health point of view. And the authors are commended for carrying it out.

An initial read showed that the abstract lacked an aim by which to appraise the paper. However, line 63 in the introduction section tells us that "this study aims to characterize the resistance, virulence and plasmid profile of Salmonella and E. coli isolated from fresh different retail meats, milk, and associated samples (handler’s hand swab, table, knife and faecal samples) in in Saboba district and Bolgatanga Municipality of Ghana" Neither the abstract or the aim reproduced above apprises the reader that the study will be using previously isolated bacteria. Mention is made in the the methods' section and 2 published papers (22 & 23) are referenced.

However, despite intense perusal of the methods section, questions arise and clarifications are thus required:

1. None of the aforementioned 2 references mentioned fresh, meat and milk samples being collected in their methods' section

2. None of these references mentioned 'lactating cows' in their methods' section or any where else in the papers.

3. None of these references mentioned (handler’s hand or milkers’ hands swab, meat sellers’ tables, knife collecting utensils, and faecal) samples being collected in their methods' section.

4. Both of these references had very similar sampling methods such that they collected 300 RTEs during almost the same period from the same (n=6) species, 50 sample per specie. But no mention is made of fresh meat, milk or tables. Why?

5. Are the authors suggesting RTEs are synonymous with fresh meat?

The authors need to clarify the above to make understanding and comprehension easy for PLoS One readers. Only then can conclusions like "This study has characterized the genomes of Salmonella and E. coli in milk, meat and their associated utensils" (line 294) have true validity as per the methods.

Reviewer #4: Review Comments

• Line 24: It is suggested to authors to use “carried mobile genetic elements” and not “carry mobile elements”.

• Line 44: Organism names must be italized

• Line 48: Authors must choose between using either Salmonella or S. enterica and keep it consistent throughout the paper.

• Line 55: It is suggested to authors to introduce “to” between “and” and “determine” to allow for a smooth read

• Line 70: How many of the isolates belonged to the various sources of isolation? This must be stated

• Line 85: In its current state, the terminal extension at 72 oC for 5 minutes appears to be part of the 35 cycles of PCR which I do not think it is. The sentence muct be revised to reflect such.

• Line 99: Do authors mean SPAdes? and was the quality of the reads assessed before assembly?

• Line 105: Multi-locus sequence types (MLST) for the isolates were determined and core-genome MLST was calculated using what tools or programs?

• Line 170: Were these just the only amr genes found considering the origin of the isolates from west Africa? Was this confirmed using other programs? I used Resfinder in Abricate (https://github.com/tseemann/abricate) and I found other genes like mdf(A) which act against macrolides in some of the isolates

• Line 226: “All multidrug resistant strains encode pO111 or IncY replicons”. What is the basics for the claim? Is it from published literature or the authors performed plasmid analysis?

• Line 246: Cite some examples of non-pathogenic ecoli for better discourse

• Line 262: It is suggested to authors to change the “possess” to "possessed"

• Line 289: “gene” should be added to “beta-lactamase”.

• Line 289: Were the ARGs identified on the plasmids or the chromosomes? This must be stated if that analysis was done?

6. PLOS authors have the option to publish the peer review history of their article (what does this mean?). If published, this will include your full peer review and any attached files.

Reviewer #1: No

Reviewer #2: **Yes: **Ayomide Adeyeye

Reviewer #3: No

Reviewer #4: **Yes: **Kingsley Emmanuel Bentum

---

## [Author Response · Author response to Decision Letter 0]

7 Nov 2024

Reviewer #1: Abstract: Review and either complete the half sentences and remove where not applicable. Eg...."While 7 of the Salmonella isolates carry the IncI1-I(Gamma) plasmid replicon."

Author’s response: Thank you for pointing this out. We have reviewed this as captured in line 22.

From Body of Manuscript (PONE-D-24-40234 (3).pdf): Suggested changing “transmission” to “disease transmission”

Author’s response: Thank you for the suggestion. We have implemented this on line 29

Review the recommendation and make it more focused on the work and not broad, blanket recommendation.

Author’s response: The recommendations have been revised and focused as requested by the reviewer. The revised recommendations are listed at the end of the conclusion (lines 325-327) 

Introduction:

The role of the equipment such knives and surfaces need to be mentioned in the introduction and not just having it appear in the objectives.

Author’s response: Thank you for this recommendation. We have included this on line 39-40 

From Body of Manuscript (PONE-D-24-40234 (3).pdf): Suggested recast for clarity “Some of these E. coli strains harbour virulence and antimicrobial resistance genes (ARGs), which is of public health concern however few have been characterized in detail”

Author’s response: Thanks for the suggestion. We have implemented this on lines 42 to 44. 

From Body of Manuscript (PONE-D-24-40234 (3).pdf): The introduction mentioned nothing about meat handling and associated surfaces. The authors may consider including this bit in the introduction.

Author’s response: Thank you for this recommendation. We have included this on line 39-40

Method:

A lot is described for Salmonella but not much is mentioned about how the E.coli samples were presented.

Author’s response: We have restructured this for more clarity. Only the salmonella isolates underwent an initial confirmation step using PCR as shown in line 82-90 after which E. coli and PCR-positive Salmonella were confirmed using VITEK as shown in line 91-93.

Results and Discussion:

I was expecting the authors to emphasize the genetic relatedness for the Salmonella and E. coli and their implication on foodborne illnesses and and control.

Author’s response: This is contained in lines 148-157 (264-283) (Salmonella) and 190-194 (E. coli) of the results section. In response to the reviewer’s comment, we have briefly discussed the significance of these findings in the discussion (lines 255-262). 

From Body of Manuscript (PONE-D-24-40234 (3).pdf): This was not described in the methods “E.coli serotyping”.

Author’s response: We have stated this in line 119. 

Other details are available in the body of the manuscript.

Overall this is an excellent work and the presentation is nicely done.

Author’s response: We thank the reviewer for the positive review and hope that all his/her concerns are addressed in the revision.

Reviewer #2: The manuscript is technically sound, with experiments conducted rigorously, including appropriate controls, replication, and sample sizes. The conclusions are well-supported by the data presented, without any overreach or misinterpretation. The statistical analysis has been performed appropriately and rigorously, ensuring that the findings are valid and reliable. The manuscript is also presented in an intelligible fashion, written in clear and standard English, making it easy to follow and understand for the intended scientific audience.

Author’s response: Thank you for your kind words and comment.

Reviewer #3: Undoubtedly, this study is of value from a public health and One Health point of view. And the authors are commended for carrying it out.

An initial read showed that the abstract lacked an aim by which to appraise the paper. However, line 63 in the introduction section tells us that "this study aims to characterize the resistance, virulence and plasmid profile of Salmonella and E. coli isolated from fresh different retail meats, milk, and associated samples (handler’s hand swab, table, knife and faecal samples) in in Saboba district and Bolgatanga Municipality of Ghana" Neither the abstract or the aim reproduced above apprises the reader that the study will be using previously isolated bacteria. Mention is made in the the methods' section and 2 published papers (22 & 23) are referenced.

Author’s response: Thank you for pointing this out. We have rephrased line 16 and 17 under abstract to portray the aim of the study as well as stating the use of previously isolated bacteria.

However, despite intense perusal of the methods section, questions arise and clarifications are thus required:

1. None of the aforementioned 2 references mentioned fresh, meat and milk samples being collected in their methods' section

2. None of these references mentioned 'lactating cows' in their methods' section or any where else in the papers.

3. None of these references mentioned (handler’s hand or milkers’ hands swab, meat sellers’ tables, knife collecting utensils, and faecal) samples being collected in their methods' section.

4. Both of these references had very similar sampling methods such that they collected 300 RTEs during almost the same period from the same (n=6) species, 50 sample per specie. But no mention is made of fresh meat, milk or tables. Why?

Author’s response: Thank you for pointing this out. We have updated the references in line 77 for the initial study that mentioned milk, milker’s hand swab, milk utensils and faecal matter from cow.

(15. Adzitey F, Yussif S, Ayamga R, Zuberu S, Addy F, Adu-Bonsu G, et al. Antimicrobial Susceptibility and Molecular Characterization of Escherichia coli Recovered from Milk and Related Samples. Microorganisms. 2022;10(7) 

 and 

27. Tibile AB. Incidence of Salmonella enterica and total bacterial count obtained from cow milk and its related samples from Saboba district: One health approach [A dissertation]: University for Development Studies, Ghana; 2022.)

5. Are the authors suggesting RTEs are synonymous with fresh meat?

Author’s response: We do not suggest that fresh meat is synonymous with RTEs. In response to the reviewer’s comment, RTE and Fresh meat are defined in line 89 of the methods section and in the key for Table 1.

The authors need to clarify the above to make understanding and comprehension easy for PLoS One readers. Only then can conclusions like "This study has characterized the genomes of Salmonella and E. coli in milk, meat and their associated utensils" (line 294) have true validity as per the methods.

Author’s response: We have clarified this as indicated above.

Reviewer #4: Review Comments

• Line 24: It is suggested to authors to use “carried mobile genetic elements” and not “carry mobile elements”.

Author’s response: Thank you for pointing this out. We have implemented this in line 25

• Line 44: Organism names must be italicized

Author’s response: We have implemented this for genus and species names in line 48 as requested and throughout the manuscript

• Line 48: Authors must choose between using either Salmonella or S. enterica and keep it consistent throughout the paper.

Author’s response: Thank you for the recommendation. We have implemented throughout as shown in lines 66, 69, 159, 241 and 324.

• Line 55: It is suggested to authors to introduce “to” between “and” and “determine” to allow for a smooth read

Author’s response: This has been added (line 60).

• Line 70: How many of the isolates belonged to the various sources of isolation? This must be stated

Author’s response: The source of each isolate is now clearly indicated on Table 1 of the revised manuscript. The sources are again indicated in Figures 1 and 2.

• Line 85: In its current state, the terminal extension at 72 oC for 5 minutes appears to be part of the 35 cycles of PCR which I do not think it is. The sentence muct be revised to reflect such.

Author’s response: We have revised the sentences as shown in lines 86-88

• Line 99: Do authors mean SPAdes? and was the quality of the reads assessed before assembly?

Author’s response: Yes, we meant SPAdes. Typo fixed in line 109. Yes, we assessed the reads quality using FASTQC as stated in the GHRU protocols.

• Line 105: Multi-locus sequence types (MLST) for the isolates were determined and core-genome MLST was calculated using what tools or programs?

Author’s response: MLST from reads were called using ARIBA as stated in line 116.

• Line 170: Were these just the only amr genes found considering the origin of the isolates from west Africa? Was this confirmed using other programs? I used Resfinder in Abricate (https://github.com/tseemann/abricate) and I found other genes like mdf(A) which act against macrolides in some of the isolates

Author’s response: We used only used AMRFinderPlus v3.10.24 for the prediction of the amr genes.

• Line 226: “All multidrug resistant strains encode pO111 or IncY replicons”. What is the basics for the claim? Is it from published literature or the authors performed plasmid analysis?

Author’s response: This was in reference to plasmid analysis performed on the E. coli isolates from our study. We have rephrased this statement as shown in line 237-238.

• Line 246: Cite some examples of non-pathogenic ecoli for better discourse

Author’s response: While most E. coli are non pathogenic, non-pathogenicity can only be verified in volunteer challenge studies. For this reason, we are unable to reliable suggest that any strains or lineages identified in this study are non-pathogens.

• Line 262: It is suggested to authors to change the “possess” to "possessed"

Author’s response: Thank you for the suggestion. This has been implemented in line 278.

• Line 289: “gene” should be added to “beta-lactamase”.

Author’s response: Thank you for pointing this out. We have implemented this in line 305.

• Line 289: Were the ARGs identified on the plasmids or the chromosomes? This must be stated if that analysis was done?

Author’s response: No we didn’t analyse for this.

(Comments in PONE-D-24-40234_reviewer_Reviewed..pdf)

Line 36: Consider using a more cautious word like "often."

Author’s response: Suggestion implemented in line 36

Line 37: To improve the flow and logical progression of ideas, it would be better to mention Salmonella before Escherichia coli in this sentence, as the subsequent sentences discuss Salmonella (salmonellosis) first. This will create a more coherent transition between the sentences.

Alternatively, to maintain consistency with sentence 2, where Escherichia coli is mentioned before Salmonella, you could reorder the following sentences to address E. coli contamination first and then discuss salmonellosis.

Author’s response: Thanks for your suggestion. We have implemented the mentioning Salmonella before E. coli as shown in lines 37, 66 and 69 

Line 38: Include more details on how S. enterica and E. coli are linked to food safety risks in Ghana beyond simply underreporting. You could cite studies from neighboring countries or similar environments to draw a broader picture.

Author’s response: We thank the reviewer for this remark. We have revised that portion of the introduction (Lines 40-42), as well as the conclusion (lines 322-327)to incorporate recent situation analyses on food safety in Ghana [Asati DA, Abdulai PM, Boateng KS, Appau AAA, Ofori LA, Agyekum TP. Food safety knowledge and practices among raw meat handlers and the microbial content of raw meat sold at Kumasi Abattoir Butchery Shops in Kumasi, Ghana. BMC Public Health. 2024 Apr 8;24(1):975. doi: 10.1186/s12889-024-18514-w. PMID: 38584288; PMCID: PMC11000319.

Botha NN, Ansah EW, Segbedzi CE, Darkwa S (2023) Public health concerns for food contamination in Ghana: A scoping review. PLoS ONE 18(8): e0288685. https://doi.org/10.1371/ journal.pone.0288685 

Botha NN, Ansah EW, Segbedzi CE, Darkwa S (2023) Public health concerns for food contamination in Ghana: A scoping review. PLoS ONE 18(8): e0288685. https://doi.org/10.1371/ journal.pone.0288685 

Christiana Cudjoe D, Balali GI, Titus OO, Osafo R, Taufiq M. Food Safety in Sub-Sahara Africa, An insight into Ghana and Nigeria. Environ Health Insights. 2022 Dec 14;16:11786302221142484. doi: 10.1177/11786302221142484. PMID: 36530486; PMCID: PMC9755555.]

Lines 40-42: Suggestion: "Some of these E. coli strains harbor virulence and antimicrobial resistance genes (ARGs), raising public health concerns. However, few of these strains have been characterized in detail."

Author’s response: Thanks for the suggestion. We have implemented this in lines 44 – 46.

Lines 46-47: The transition to the second sentence is somewhat abrupt. While it maintains the focus on Salmonella and E. coli, it shifts to discussing antibiotic resistance without explicitly connecting this to the public health implications mentioned in the first sentence. 

Adding a linking phrase like "In addition to contamination risks" or "Another major concern" would help connect contamination and resistance issues.

Author’s response: Thanks for the suggestion. We have implemented this in lines 49- 50.

Line 66: Overall, the literature review does an adequate job of introducing the relevant keywords and concepts. However, to strengthen its coherence and logical flow, the introduction could provide a deeper review of S. enterica and E. coli in Ghana, offer more regional context regarding food safety and antimicrobial resistance, and provide clearer transitions between general and Ghana-specific trends. Additionally, ensure that all abbreviations (e.g., ARGs, MDR) are defined at first use.

Author’s response: We have expanded the introduction and cited additional references on Salmonella enterica and E. coli in Ghana in lines 42-44.

Line 69: This section clearly describes the source of the bacterial isolates (Salmonella and E. coli) used in the study and their storage. It adequately connects to the objective of studying these specific bacteria from meat, milk, and related environments.

Author’s response: Thanks for this

Line 133: Overall, the methods section provides a clear and detailed description of the techniques used to achieve the study's objectives. However, it could be improved by:

1. Providing a brief rationale for why certain tools or databases were chosen over others. (e.g. SISTR was used or Salmonella serotyping due to its high accuracy and ability to predict serovar from whole genome sequencing data, which aligns with our genomic approach.

2. Link methods to objectives. (For example: "To identify antimicrobial resistance genes and predict potential resistance phenotypes, crucial for our resistance profiling objective, we employed AMRFinderPlus v3.10.24..."]

Author’s response: While we have compared the tools we use to others and selected those that perform well on West African isolates for this study, we did not conduct a systematic evaluation of comparable tools for this study. For this reason, we are hesistant to recommend the tools we used above others. In response to the reviewer’s concern, we have cited the references that emanated from our earlier evaluation and stated the purpose for each tool choice on lines 114-115. 

 Line 168: Good! This corresponds to the objectives of characterizing plasmid and resistance profiles

Author’s response: Thanks for this

Line 227: While the results are presented in a logical order, they don't strictly follow a chronological order based on the objectives as typically stated in the introduction section. A more chronological order based on typical objectives might be:

Isolate Identification and Characterization-(Serotypes and Sequence Types)

Antibiotic Resistance Gene Profiles

Virulence Factor Profiles

Plasmid Profiles

Author’s response: Similar methods were performed for the most part for E. coli and Salmonella and they are mostly presented, for each species, in the order the reviewer recommends. However, we did not present the findings in a listed manner in order to permit a synthetic reading of the data, which is the state of the art for genomic data sets [Argimón S, Abudahab K, Goater RJE, Fedosejev A, Bhai J, Glasner C, Feil EJ, Holden MTG, Yeats CA, Gr

---

## [Decision Letter · Decision Letter 1]

26 Nov 2024

PONE-D-24-40234R1Genomic characterization of foodborne Salmonella enterica and Escherichia coli isolates from Saboba district and Bolgatanga Municipality GhanaPLOS ONE

Dear Dr. Okeke,

Thank you for submitting your manuscript to PLOS ONE. After careful consideration, we feel that it has merit but does not fully meet PLOS ONE’s publication criteria as it currently stands. Therefore, we invite you to submit a revised version of the manuscript that addresses the points raised during the review process.

We look forward to receiving your revised manuscript.

Kind regards,

Mabel Kamweli Aworh, DVM, MPH, PhD. FCVSN

Academic Editor

PLOS ONE

Journal Requirements:

Reviewers' comments:

Reviewer's Responses to Questions

**Comments to the Author**

1. If the authors have adequately addressed your comments raised in a previous round of review and you feel that this manuscript is now acceptable for publication, you may indicate that here to bypass the “Comments to the Author” section, enter your conflict of interest statement in the “Confidential to Editor” section, and submit your "Accept" recommendation.

Reviewer #1: All comments have been addressed

Reviewer #2: All comments have been addressed

Reviewer #3: All comments have been addressed

Reviewer #4: (No Response)

2. Is the manuscript technically sound, and do the data support the conclusions?

Reviewer #1: Yes

Reviewer #2: Yes

Reviewer #3: Yes

Reviewer #4: Yes

3. Has the statistical analysis been performed appropriately and rigorously? 

Reviewer #1: N/A

Reviewer #2: Yes

Reviewer #3: N/A

Reviewer #4: N/A

4. Have the authors made all data underlying the findings in their manuscript fully available?

Reviewer #1: (No Response)

Reviewer #2: Yes

Reviewer #3: Yes

Reviewer #4: Yes

5. Is the manuscript presented in an intelligible fashion and written in standard English?

Reviewer #1: Yes

Reviewer #2: Yes

Reviewer #3: Yes

Reviewer #4: Yes

6. Review Comments to the Author

Reviewer #1: Great work. The authors have addressed all the comments from the initial submission. There are no comments from me at this time.

Reviewer #2: (No Response)

Reviewer #3: (No Response)

Reviewer #4: • Line 25: Authors must understand that research publication is a reported work done hence all verbs must be in the past tense. Hence “belong” must be “belonged” and “carrying” be “carried”. The manuscript is rife with such errors and must be corrected

• Line 25: I again ask the authors to define which mobile elements as there are several. If it is a plasmid, then authors must state it.

• Line 98-99: This implies that apart from the VITEK no initial confirmation was done for the E. coli strains like Salmonella. Any reason for that?

• Line 128: Please provide a reference or link for the VFDB platform or publication

• Line 183: Same comment as Line 25

• Line 190: The “,” between “other” and “tetA” is not needed.

• Line 191-192: The plasmid information is conspicuously missing in the Figure 1

• Line 196: I suggest the addition of sequence type to the title like in Salmonella

• Line 294- 296: The sentence is not reading right. Authors must rephrase it

• Line 303: It is suggested to authors to be more specific by writing “ association of the Inc1 plasmid replicon

• Line 313: Same comment as in Line 25

• Line 326: Should read “multidrug-resistant”

• Line 330-331: The sentence is not reading right

• Line 333: Saying “elsewhere in West Africa” is vague. Rather authors should say “and by extension West Africa”.

7. PLOS authors have the option to publish the peer review history of their article (what does this mean?). If published, this will include your full peer review and any attached files.

Reviewer #1: No

Reviewer #2: No

Reviewer #3: No

Reviewer #4: No

---

## [Author Response · Author response to Decision Letter 1]

27 Nov 2024

Reviewer #4: 

• Line 25: Authors must understand that research publication is a reported work done hence all verbs must be in the past tense. Hence “belong” must be “belonged” and “carrying” be “carried”. The manuscript is rife with such errors and must be corrected

Author’s response: The reviewer is correct that the work was done in the past, but the strains still carry the plasmids mentioned in the paper, which is why past tense was not used. However to satisfy the reviewer, we have changed this in line 22 and 25

• Line 25: I again ask the authors to define which mobile elements as there are several. If it is a plasmid, then authors must state it.

Author’s response: We have revised this statement in line 25.

• Line 98-99: This implies that apart from the VITEK no initial confirmation was done for the E. coli strains like Salmonella. Any reason for that?

Author’s response: All the isolates were ultimately sequenced, which is the best confirmation we could have had. We used other methods to preview isolates for sequencing. E. coli is relatively simple to assure using basic cultural methods. Salmonella, a bit more challenging, which is why we used VITEK.

• Line 128: Please provide a reference or link for the VFDB platform or publication

Author’s response: We have added the link for the VFDB in line 128

• Line 183: Same comment as Line 25

Author’s response: We have implemented this in lines 161, 163 and 183 as recommended.

• Line 190: The “,” between “other” and “tetA” is not needed.

Author’s response: We have removed the “,” as suggested in line 190.

• Line 191-192: The plasmid information is conspicuously missing in the Figure 1

Author’s response: We have reconstructed figure 1 to show the plasmid information in line 173 and 177.

• Line 196: I suggest the addition of sequence type to the title like in Salmonella

Author’s response: We have added this in line 196.

• Line 294- 296: The sentence is not reading right. Authors must rephrase it

Author’s response: We have rephrased this as captured in line 293-297

• Line 303: It is suggested to authors to be more specific by writing “ association of the Inc1 plasmid replicon

Author’s response: We have rephrased this as captured in line 304

• Line 313: Same comment as in Line 25

Author’s response: We have implemented this in line 314 and 327

• Line 326: Should read “multidrug-resistant”

Author’s response: We have implemented this in line 327

• Line 330-331: The sentence is not reading right

Author’s response: We added a comma after regulations in line 330 and changed “them” to “regulations” for a smooth read. 

• Line 333: Saying “elsewhere in West Africa” is vague. Rather authors should say “and by extension West Africa”.

Author’s response: We have implemented this in line 333 and 334

---

## [Editor Report · Decision Letter 2]

28 Nov 2024

Genomic characterization of foodborne Salmonella enterica and Escherichia coli isolates from Saboba district and Bolgatanga Municipality Ghana

PONE-D-24-40234R2

Dear Dr. Okeke,

We’re pleased to inform you that your manuscript has been judged scientifically suitable for publication and will be formally accepted for publication once it meets all outstanding technical requirements.

Kind regards,

Mabel Kamweli Aworh, DVM, MPH, PhD. FCVSN

Academic Editor

PLOS ONE
---

## [Editor Report · Acceptance letter]

5 Dec 2024

PONE-D-24-40234R2 

PLOS ONE

Dear Dr. Okeke, 

I'm pleased to inform you that your manuscript has been deemed suitable for publication in PLOS ONE. Congratulations! Your manuscript is now being handed over to our production team.

Kind regards, 

on behalf of

Dr. Mabel Kamweli Aworh 

Academic Editor

PLOS ONE